# Self-Reported Dyslexia Traits as Positive Predictors of Self-Reported Cognitive Failures in the Workplace

**DOI:** 10.3390/bs15111582

**Published:** 2025-11-18

**Authors:** James H. Smith-Spark, Madalyn Huang

**Affiliations:** 1School of Allied Health and Life Sciences, London South Bank University, 103 Borough Road, London SE1 0AA, UK; mh6730@nyu.edu; 2Steinhardt School of Culture, Education, and Human Development, New York University, New York, NY 10012, USA

**Keywords:** dyslexia, neurodiversity, ADHD, cognition, cognitive failures, workplace, employment

## Abstract

There is little direct empirical evidence indicating how dyslexia-related cognitive difficulties express themselves in employment settings, although employers may be legally required to support neurodivergent workers through targeted accommodations. The current pre-registered online survey investigated the relationship between dyslexia traits and the self-reported frequency of workplace cognitive failures. Four hundred native English speakers were recruited via Prolific. All respondents identified as being full- or part-time UK employees in a central place of work. The respondents completed a series of published self-report questionnaires on neurodivergent symptomatology, mental wellbeing, personality characteristics, and busyness and routine at work. Hierarchical multiple regression analyses were used to determine whether dyslexia traits were predictive of workplace cognitive failures. After controlling for age, busyness and routine, Big Five personality traits, mental wellbeing, and self-reported ADHD symptoms, dyslexia traits were a significant positive predictor of the overall frequency of workplace cognitive failures. Self-reported dyslexia traits were also significant positive but weak predictors of the individual memory, attention, and action factors. The predictive relationships found between self-reported dyslexia traits and workplace cognitive failures suggest that a similar approach with officially diagnosed individuals would prove fruitful in understanding how dyslexia affects work performance and improve targeted support for dyslexic employees.

## 1. Introduction

There is a growing focus on understanding and supporting neurodiversity in the workplace (e.g., [19]; [36]; [50]; [60]). Developmental dyslexia, one such neurodivergent condition, is estimated to affect 8% of working age adults ([58]), with the [87] ([87]) estimating that 6.2 million adults of working age are dyslexic in the UK based on 2014 statistics. Moreover, a disproportionate percentage of adults with dyslexia estimated to be unemployed has been reported ([87]; [32]). Despite these concerning statistics and an internationally recognised need to inform job selection procedures, workplace interventions, and line management practices and organisational procedures (e.g., [15]; [80]; [84]), very little research has been conducted into the impact of dyslexia on workplace cognition (although see [63], for evidence of specific dyslexia-related deficits in a virtual reality office-based environment). This gap in the literature stands in contrast to the corpus of research into the psychosocial impact of dyslexia on work (see [16], [17], for systematic reviews) and indications that the cognitive demands placed on dyslexic workers can lead to employee burnout ([85]). The current study was aimed at addressing the dearth of literature on dyslexia and cognition in employment settings, focusing specifically on the relationship between dyslexia traits and workplace cognitive failures in a sample of UK workers recruited via the Prolific participant recruitment and data collection platform.

Dyslexia is typically characterized as affecting the processing of phonological information and the development of reading and spelling processes (see recent considerations by [70]; [78]). In addition to these core phonological processing deficits (see [79]; [11], for reviews), there is also a growing corpus of evidence demonstrating that dyslexia also has a broader negative impact on cognitive functioning (for meta-analytic reviews, see [6]; [41]). Such effects continue to be found in adulthood, with dyslexia-related difficulties having been found under laboratory conditions in, for example, memory (e.g., [55]; [61], [67]) and executive function (e.g., [9]; [64], [63]). Laboratory-based research of this nature provides important insights into the effects of dyslexia on cognition and, further to this, self-report questionnaires and interviews give a general indication of its effects in day-to-day life (e.g., [64], [67]; [65]). However, less is known about the relationship between dyslexia (or dyslexia traits) and cognition in the specific everyday contexts in which adults need to function successfully. One such specific context is the workplace, the focus of the current study, which utilizes a community sample approach as an initial means of understanding the ways in which dyslexia traits can influence cognition in employment settings.

Cognitive failures reflect all conceivable types of failure of memory, language, and attention in everyday life (e.g., [77]; for a systematic review, see [10]). A common way to assess the frequency with which such cognitive failures are experienced has been through the self-report Cognitive Failures Questionnaire (CFQ; [7]; although see recent critiques by [26], [26]; [28]. The CFQ has been used in dyslexia research. [61] ([61]) found significant group differences on the CFQ, with officially diagnosed dyslexic university students reporting themselves to experience more frequent cognitive failures than non-dyslexic students. These group differences were corroborated by proxy ratings gained from close associates of the CFQ respondents, with the dyslexic respondents being rated as significantly more prone to cognitive failures by housemates, partners or family members, with a moderate positive correlation being found between self- and proxy-ratings. In a sample of adult dyslexics (88% of whom were employed and an additional 6% were self-employed), [38] ([38]) found a weak to moderate negative relationship between self-reported levels of work-based self-efficacy and cognitive failures as measured by the CFQ.

However, [81] ([81]) have highlighted the limitation of the CFQ ([7]) in treating cognitive failures as an individual differences trait which is expressed across a range of different everyday contexts and not specifically within the work context. In consequence, they developed the Workplace Cognitive Failure Scale (WCFS; [81]) as a work-specific measure focused on predicting workplace behaviour. The WCFS was validated based on responses from full-time US navy personnel and USA-based employees in, for example, production, construction, and plant operations. Using factor analysis, Wallace and Chen identified three factors underlying responses to the WCFS. These factors were Memory, Attention, and Action. The Memory factor relates to failures to retrieve relevant and familiar work-related information. The Attention factor reflects failures to retain attentional focus on task-relevant work information. Finally, the Action factor reflects the failure to carry out appropriate actions or behaviours as intended while working. Research using the WCFS has included, for example, studies on nurses ([3]; [21]; [30]; [31]), teachers ([35]), work-family conflict ([37]) and risk-taking in after-work sports activities ([22]).

Given that little empirical evidence exists concerning the relationship between dyslexia and cognition in the workplace, the current study was conducted to investigate whether there was an association between dyslexia symptomatology and the incidence of cognitive failures at work, given that this is an important facet of effective workplace performance (e.g., [81]) and one where technological support can potentially play a role in reducing errors (e.g., [13]; [14]). In the current study, a similar online survey approach to that used by [54] ([54]) was employed, involving a UK sample of Prolific users in full- or part-time work across a range of professions. The 15-item self-report Adult Reading Questionnaire (ARQ; [69]) was used to determine the severity and frequency of reading difficulties consistent with dyslexia through measurements of literacy, word-finding, language, and organization. Part A of the Adult ADHD Self-Report Scale (ASRS; [33]) evaluated symptoms consistent with a diagnosis of ADHD. The Workplace Cognitive Failure Scale (WCFS; [81]) was presented to determine the frequency of cognitive failures within the workplace through measures of action, memory, and attention. While the focus of th study was on the relationship between dyslexia traits and workplace cognition, other contributors to workplace cognitive failures have been identified in the self-report literature and, thus, needed to be controlled to rule them out as alternative explanations for any predictive associations between self-reported dyslexia traits and workplace cognitive failures. These additional contributors will now be considered in turn.

To control statistically for individual differences in busyness (density of obligations) and routine in daily life (predictability of events), the Martin and Park Environmental Demands Questionnaire was administered (MPED; [43]). Busyness was hypothesized to be a positive predictor of workplace cognitive failures, with [75] ([75]) reporting positive relationships between Busyness and both retrospective and prospective memory failure in general everyday contexts, while Routine was likely to show less of a relationship, with this prediction again being based on prospective memory in general daily life (e.g., [44]; [75]).

The 10-item Big Five Inventory (BFI-10; [56]) was also presented as a further statistical control, since personality traits (particularly neuroticism and conscientiousness) have been associated with cognitive failures and work performance ([81]). Based on Wallace and Chen’s findings, neuroticism was expected to be a positive predictor of workplace cognitive failures, while conscientiousness was predicted to have a negative relationship with workplace cognitive failures.

The Short Warwick-Edinburgh Mental Well-being Scale (SWEMWBS; [73]) was used as a statistical control for mental wellbeing, focusing on its functional aspects. Given the documented links between mental health concerns and cognitive failures (e.g., [45]; [42]; [51]; [74]), negative predictive relationships between mental wellbeing scores and workplace cognitive failures were expected.

Based on previous self-report research on cognitive failures in adults with official diagnoses of dyslexia ([61]), it was predicted that higher self-reported incidences of cognitive failure in the workplace would be reported by workers with higher self-reported symptoms of dyslexia and that this predictive relationship would be found after controlling for ADHD symptomatology, busyness at work, mental wellbeing, and personality type. While the current study focused on self-reported dyslexia traits in a community sample rather than testing officially diagnosed individuals, it seemed plausible to generate hypotheses for each of the three WCFS factors (cognitive failures relating to Memory, Attention, and Action) based on the research literature on dyslexia. A predictive relationship between dyslexia traits and the WCFS Memory component was also expected, based on research investigating various memory systems in adults with formal diagnoses of dyslexia. Such evidence comes from self-reports into different types of everyday memory (e.g., [54]; [61]; [65]; [66]), objective laboratory tests of memory (e.g., [48]; [55]), for some memory systems, a combination of the two approaches (e.g., [67]). For the WCFS Attention component, heightened dyslexia traits were also expected to predict greater self-reported attentional difficulties in the workplace. In a community sample, [54] ([54]) found that increased levels of self-reported dyslexia traits were predictive of greater and more frequent self-reported attentional difficulties across different everyday contexts. A similar relationship was expected for perceived attentional problems in the workplace. It was less certain whether a predictive relationship between dyslexia traits and the WCFS Actions component would be found. For officially diagnosed university students, [61] ([61]), using [7]’s ([7]) CFQ, found group differences on some, but not all, items which asked about the frequency of executing actions that were unintended (using [53], factor structure of the CFQ). Further to this finding, experimental evidence of non-significant differences in action-based prospective memory in diagnosed dyslexic adults ([63]) also suggested that the relationship between dyslexia traits and Actions might be of a smaller magnitude than that obtained for either the WCFS Memory or the WCFS Attention component.

Positive predictive relationships between self-reported ADHD symptoms and workplace cognitive failures were expected. This prediction was based, firstly, on [24]’s ([24]) analysis of how the cognitive profile of ADHD is likely to play out in the workplace and, secondly, on adults with diagnosed ADHD self-reporting more frequent general cognitive failures on [7]’s ([7]) CFQ, (e.g., [29]; [34]; [40]; [86]).

The current online self-report survey, therefore, sought to determine whether adult dyslexia traits would predict the frequency with which workplace cognitive failures were experienced by workers who always worked in a central place of employment, after controlling for age, busyness and routine in everyday life, Big Five personality characteristics, mental wellbeing, and co-occurring ADHD symptoms.

## 2. Materials and Methods

### 2.1. Participants

The respondents were recruited via the online participant recruitment platform Prolific and received a payment of GBP 3 in appreciation of their time. For budgetary reasons, the sample size was limited to 400 eligible respondents based on their answers to initial pre-screening questions that were verified against their Prolific profiles. The respondents were required to state that they were aged between 18 and 66 years (with the latter being the current state pension age in the UK), were in current full- or part-time employment in the UK, that they always worked from a central place of work, and that English was their first language. In total, 467 responses were made, of which 67 were ineligible due to their answers to these pre-screener questions and their participation was terminated by the Qualtrics survey after responding to those initial questions. Of the final sample of 400 respondents meeting the pre-screening criteria (out of a total of 7367 matching participants on Prolific), 232 identified as female, 167 as male, and one as non-binary. In terms of ethnic group, 13 respondents identified as being Asian, 27 as Black, 11 as Mixed, and 349 as White. The mean age of the sample was 39.89 years (*SD* = 12.07). There were 279 respondents who indicated that they worked full-time and a further 121 who indicated that they worked part-time. Overall, the respondents reported that they usually worked for a mean 33.47 h (*SD* = 10.53) per week. For the full-time respondents, the mean hours reported usually being worked was 39.26 h per week (*SD* = 5.01) and for the part-time respondents, the mean was 20.12 h (*SD* = 7.30) per week. A participants pyramid providing further details on the sample characteristics is presented in Figure 1.

### 2.2. Materials

The ARQ ([69]) is a 15-item self-report questionnaire that evaluates adult reading difficulties through measures of literacy, language, and organization. Six items (ARQ 5–10) asked participants to rate their personal dyslexia-related tendencies in literacy, word-finding, and organization (scored on scale 0–4; 0 = Never, 1 = Rarely, 2 = Sometimes, 3 = Frequently, and 4 = Always). Two items asked participants about their frequency of reading and writing in everyday life (ARQ 4 & 11). For these two items, the scale was reversed (where 4 = Never, 3 = Rarely, 2 = Sometimes, 1 = Frequently, and 0 = Always). Item 12 directly related to the preceding items and used a three-point scale (where 1 = Yes, 0 = No, and 0.5 = Maybe). Items 1 and 2 used a three-point scale (where 0 = Yes, 1 = No, 0.5 = Don’t know). Item 3 used a four-point scale (scored 0–3; 0 = Good, 1 = Average, 2 = Poor, 3 = Very poor). Item 13 prompted the participant to appraise their difficulties on a four-point scale (scored 0–3; 0 = Good, 1 = Average, 2 = Poor, and 3 = Very Poor). The last two items (ARQ 14 & 15) asked the participant about previous diagnosis on a two-point scale (where 1 = Yes and 0 = No). If the participant indicated that they had been previously diagnosed with dyslexia, they were then prompted to indicate by whom they were assessed (e.g., an educational psychologist). The total ARQ score ranges from 0–43 with a higher score indicating increased likelihood of impairment relating to literacy, language, and word organization ([69]).

Part A of the ASRS ([33]) screener is a six-question screener that evaluates participants based on the self-reported frequency of DSM-IV Criterion A symptoms of adult ADHD that they exhibit ([1]). Four items (ASRS 1–4) investigated difficulties with sustained attention, organization, and prospective memory and two items (ASRS 5 and 6) investigated difficulties related to hyperactivity. In this study, since severity of symptoms was not relevant to analysis, the questions were scored on a binary scale. All items were scored on a five-point scale (where 0 = Never, 1 = Rarely, 2 = Sometimes, 3 = Often, and 4 = Very Often). The total ASRS score ranged from 0–24 with a higher score indicating that the participant reported higher rates of attention deficits, organization challenges, and hyperactivity ([33]).

The WCFS ([81]) is a 15-item self-report questionnaire that evaluates cognitive failures in the workplace context. Wallace and Chen reported a three-factor structure, in which five items (WCFS 1–5) measure memory, five items (WCFS 6–10) assess attention, and five items (WCFS 11–15) probe action. All items were scored on a five-point scale (where 1 = Never, 2 = Rarely, 3 = Sometimes, 4 = Often, and 5 = Very Often). The total WCFS score ranges from 15–75 with higher scores indicating higher self-reported frequencies of cognitive failure in the workplace ([81]).

The selection of self-report measures to control statistically for other contributors to cognitive failure were made on the dual basis of a good pedigree in the literature and, more pragmatically (where different options presented themselves as instruments), a smaller number of items in order to retain the goodwill of respondents and ensure the quality of responses as far as possible by not creating a survey that was overly long.

The MPED ([43]) is an 11-item self-report questionnaire that measures a participant’s levels of daily environmental demands broken down into measures of Busyness and Routine, each assessed on a five-point scale. The first item was scored 1 = Not busy at all, 2 = Rarely, 3 = Somewhat Busy, 4 = Very Busy, and 5 = Extremely Busy. All remaining 10 items were scored 1 = Never, 2 = Rarely, 3 = Sometimes, 4 = Often, and 5 = Very Often. The MPED yields two scores, a Busyness score and Routine score. The first seven items measure busyness, while the final four items measure routine. The Busyness score relates to the pace or density of events in daily life to which the individual has to respond and represents the sum of the responses to the first seven items. Busyness scores ranged from a minimum of seven to a maximum of 35, with higher scores in this section indicating higher levels of self-reported busyness. The Routine score reflects the predictability or routinized nature of daily events independent of Busyness and is the sum of the final four items. Routine scores ranged from four to 20, with higher scores in this section indicating higher levels of perceived routine in day-to-day life ([43]).

The BFI-10 ([56]) is a 10-item self-report questionnaire that measures personality in terms of extraversion, agreeableness, conscientiousness, neuroticism, and openness. For every dimension of personality measured, there were two questions, one that was true-scored (where 1 = disagree strongly, 2 = disagree a little, 3 = neither agree not disagree, 4 = agree a little, 5 = agree strongly) and one that was false-scored (5 = disagree strongly, 4 = disagree a little, 3 = neither agree nor disagree, 2 = agree a little, 1 = agree strongly). For each personality category, the score ranged from 2–10 with higher scores indicating that the trait is more apparent. Extraversion indicates a tendency towards excitement-seeking, gregariousness, and assertiveness. Agreeableness indicates a tendency towards trust, altruism, and straightforwardness. Conscientiousness indicates an inclination towards competence, self-discipline, and achievement striving. Neuroticism indicates a tendency towards anxiety, hostility, and self-consciousness. Lastly, openness indicates a tendency to be curious, imaginative, and artistic (BFI-10; [56]).

The SWEMWBS ([73]) is a seven-item self-report measure of mental wellbeing that focuses on the functional aspects of mental wellbeing rather than feelings (see [59], for an investigation into its construct validity with a clinical sample). Participants indicated their response to each question on a five-point scale from 1 to 5. The minimum score was, thus, seven and the maximum score of 35, with higher scores indicating higher levels of positive mental wellbeing.

### 2.3. Design

To check for multicollinearity, bivariate Pearson’s product-moment correlations were performed on all predictor variables entered into the hierarchical regression analyses.

Separate multiple hierarchical regression analyses were run on WCFS total score and the Memory, Attention, and Action WCFS factors. In each case, the predictor variables were added to the model in the following order: Block 1: self-reported age, Block 2: MPED Busyness and Routine scores, Block 3: BFI-10 scores for Extraversion, Agreeableness, Conscientiousness, Neuroticism, and Openness, Block 4: SWEMWBS scores, Block 5: ASRS scores, and Block 6: ARQ score.

This study was pre-registered with the Open Science Framework (https://doi.org/10.17605/OSF.IO/CTNSK).

### 2.4. Procedure

Full ethical approval was granted by the first author’s host institution (School of Applied Sciences Ethics Committee, reference number ETH2324-0176). After following the link from Prolific to the survey, respondents gave informed consent to take part. After this, they responded to questions about their age and the job sector in which they worked (selecting their choice from a list). The respondents then answered (in order of presentation) the ARQ ([69]), the ASRS ([33]), the WCFS ([81]), the MPED ([43]), the BFI-10 ([56]), and the SWEMWBS ([73]). On the final page of the survey, a written debrief explaining the nature of this study and the expected findings was presented. The survey was published on Prolific on 12th March 2025 and was closed on 20th March 2025 after the target N of 400 eligible respondents was achieved.

## 3. Results

### 3.1. Demographic Characteristics

Table 1 shows the job sectors in which the respondents identified as working. Healthcare, Education, and Business Administration were the most frequently identified, constituting between them nearly three quarters of the job sectors stated in the list provided to respondents (with 41% of the sample indicating “Other”).

### 3.2. Descriptive Statistics and Correlations

The descriptive statistics and Cronbach’s α-values for the questionnaire-based predictor and outcome variables are presented in Table 2.

The Pearson’s product-moment correlations are presented in the Appendix A. The magnitude of the Pearson’s correlations (all *r* ≤ ±0.631) indicated that, while some measures were significantly intercorrelated, there were no concerns over strong multicollinearity. Given the detailed analysis plan set out in the OFS pre-registration of this study (which identified in which block each variable would be entered into the analyses), all the predictor variables were entered into the hierarchical regression analyses at the specified point regardless of whether they correlated with the corresponding output variable.

Following these checks for reliability and multicollinearity, hierarchical multiple regressions were performed on WCFS total score and scores on the three WCFS factors.

### 3.3. Total WCFS Score

The final hierarchical multiple regression model accounted for 66.7% of the variance in total WCFS score (adjusted-*R*^2^), with the addition of ARQ scores in Block 6 accounting for a highly significant (*p* < 0.001) extra 3.5% of the variance being explained. The combination of the predictor variables was a highly significant predictor of total WCFS scores, *F*(11,388) = 73.59, *p* < 0.001. Of the predictor variables, five were found to be significant independent predictors of WCFS scores. Scores on the ASRS were a moderate positive predictor, while ARQ scores, Busyness, and Routine were all weak positive predictors of workplace cognitive failures. Conscientiousness scores were weak negative predictors. No other predictor variables showed significant predictive relationships with WCFS scores. Table 3 shows the test statistics for the final hierarchical regression model).

### 3.4. The WCFS Memory Factor

In the case of the WCFS Memory factor, 57.7% of the variance (adjusted-*R*^2^) was explained by the combination of the predictor variables in the final model. An additional 3.7% of the variance was explained by entering ARQ scores in Block 6, a highly significant increase (*p* < 0.001). The final hierarchical regression model was a highly significant predictor of self-reported workplace failures of memory, *F*(11,388) = 50.54, *p* < 0.001. Of the individual predictor variables, highly significant weak positive predictive relationships with the WCFS Memory factor were found for Busyness and ARQ scores, while a highly significant moderate positive relationship was found in the case of ASRS scores. Conscientiousness had a significant weak negative relationship with WCFS memory factor scores. The test statistics for the WCFS Memory factor are shown in Table 4.

### 3.5. The WCFS Attention Factor

Of the variance in WCFS Attention scores, 56.9% (adjusted-*R*^2^) was accounted for by the final hierarchical regression model. The final model was a highly significant predictor of scores on the WCFS Attention factor, *F*(11,388) = 48.96, *p* < 0.001. When ARQ scores were entered into the model in Block 6, an additional 2.2% of the variance was explained and this *R*^2^ change was highly significant (*p* < 0.001). Of the individual predictor variables, significant to highly significant weak predictive relationships were found between participant age, Busyness, Routine, and ARQ scores and scores on the WCFS Attention factor, while ASRS scores had a highly significant moderate positive relationship with the WCFS Attention factor. Finally, a significant weak negative relationship was found between Conscientiousness and the WCFS Attention factor. Table 5 shows the regression statistics for the WCFS Attention factor.

### 3.6. The WCFS Action Factor

The final hierarchical regression model significantly predicted scores on the WCFS Action factor, *F*(11,388) = 25.07, *p* < 0.001, accounting for 39.9% (adjusted-*R*^2^) of the variance. A highly significant 2.3% of additional variance was explained by entering ARQ scores into the final model. When considering the contributions of the individual predictor variables to the final overall model, Busyness, Neuroticism, ASRS scores, and ARQ scores were all found to be significant to highly significant weak positive predictors of scores on the WCFS Action factor. A significant weak negative relationship was also found between Conscientiousness and WCFS Action factor scores. The hierarchical regression test statistics for the WCFS Action factor are presented in Table 6.

## 4. Discussion

The current online self-report study explored the relationship between dyslexia traits and workplace cognitive failures in a UK-based sample of 400 Prolific users who identified themselves as being in full- or part-time employment, working in a central work location, and having English as their first language. Overall, dyslexia traits (as measured by the ARQ; [69]) were significant positive predictors of the self-reported frequency of workplace cognitive failures (as measured by the WCFS; [81]). This predictive relationship was found after controlling for participant age, busyness and routine, Big Five personality characteristics, mental wellbeing, and ADHD symptomatology. In the case of the three WCFS factors (Memory, Attention, and Action), dyslexia traits accounted for an additional (and highly statistically significant) 2.2 to 3.7% of the variance in self-reported workplace cognitive failures within the hierarchical regression models. Dyslexia traits explained more unique variance for the Memory factor, with the Action and Attention factors explaining approximately equal amounts of the variance. Across all three WCFS factors, the strength of the relationships (as indicated by the standardized-β values reported in Section 3.3, Section 3.4, Section 3.5 and Section 3.6) between self-reported dyslexia traits and workplace cognitive failures was lower than it was for self-reported ADHD symptoms but was, nonetheless, highly significant for each WCFS factor. This pattern of results would indicate that, although higher ADHD symptoms are associated with greater levels of cognitive challenge in the workplace (and are themselves consistent with a more general documented increased susceptibility to everyday cognitive failures in adults with a confirmed diagnosis of ADHD; e.g., [29]; [34]; [40]; [86]), elevated levels of dyslexia traits are also (and independently) linked to a greater propensity to cognitive failure at work.

In finding a positive predictive relationship between dyslexia traits and workplace cognitive failures, the findings of the current study are consistent with previous work highlighting an increased general vulnerability to cognitive failure in adults with an official diagnosis of dyslexia ([38]; [61]). The present results thus add to the corpus of research on dyslexia-related cognitive failure and also extend this literature by indicating how this relationship is likely to manifest itself in a specific and very important facet of adult life, namely employment. As noted in Section 1, dyslexia-related impairments in memory have been documented across different memory systems under both laboratory conditions in officially diagnosed dyslexics ([48]; [55]; [67]) and through self-reports of everyday difficulties in both diagnosed dyslexics and a community sample (e.g., [54]; [65]; [66], [67]). The greater association found in the current study between dyslexia traits and workplace cognitive failures involving memory may reflect a compounding effect of phonological processing difficulties and flexible access to information in long-term memory at the time that it is required (see [64], for a consideration of these dual demands on phonological processing and executive function).

On a broader theoretical level (and assuming similar patterns are found in workers with an official diagnosis of dyslexia), the findings also highlight the need for dyslexia theories to accommodate broader cognitive difficulties that extend beyond reading and spelling in their explanatory accounts. The results of the current study indicate that dyslexia traits are related to an important cognitive aspect of workplace performance and need to be explained by theory in order for increased difficulties in this area to be understood and addressed through support. The dyslexia theory best able to explain the current findings is the Dyslexia Automatisation Deficit hypothesis ([46]; see also [47]). This hypothesis argues, as a result of failing to fully automatise performance of cognitive and motor skills, dyslexics need to consciously compensate for this lack of automaticity by allocating extra attentional resources to their perform the skill. This conscious effort can result in slower and more error-prone performance, particularly when having to perform more than one task at the same time. [62] ([62]) have also addressed dyslexia-related everyday difficulties in terms of executive (consciously controlled) and automatic processes.

The reasons why self-reported ADHD symptoms are more greatly associated with workplace cognitive failures than self-reported dyslexia traits could be explored in future work using objective measures of work performance (using simulated or virtual reality tasks) and accompanying neuropsychological measures of cognitive load (see [39]) to gain greater insights into how work-related cognition is differently affected by the two neurodevelopmental conditions.

Cognitive abilities aside, conscientiousness is considered to be the most powerful predictor of workplace performance ([83]). This argument is certainly borne out by the results of the current study, with Conscientiousness having significant negative predictive relationships with workplace cognitive failures, in terms of WCFS total score and all three WCFS factors. The only other Big Five (as measured by the BFI-10; [56]) personality characteristic to show a significant (but in this case positive) relationship with workplace cognitive failures was neuroticism. In all these cases, personality characteristics had only weak relationships with workplace cognitive failure and the general pattern is consistent with the findings of [81] ([81]), who also reported conscientiousness and neuroticism as being the personality characteristics most strongly related to self-reported workplace cognitive failures. Low conscientiousness has also been found to be related to more workplace accidents ([12]; [82]). Furthermore, systematic reviews and meta-analyses that have examined the associations between personality characteristics and everyday cognitive failures more generally have also highlighted similar relationships with conscientiousness and neuroticism ([4]; [74]). To explain the negative relationship between conscientiousness and cognitive failure, [74] ([74]) argue that people reporting high levels of conscientiousness also tend to express this personality trait in their behaviour, being more organised in their approach to their possessions and to their schedules (see [4], for a similar argument). As a result, such individuals are better able to pay attention ([74]) or have a more structured mindset (Aschwanden et al.) and thus avoid cognitive failures arising from blunders and distraction. In the case of the positive relationship between neuroticism and workplace cognitive failures, Aschwanden et al. present a “mental processes model” which argues that people high in particular personality traits have characteristic mental processes that make them either more susceptible to or more resilient against cognitive failures. They argue that neuroticism is associated with rumination which may lead to distraction but also that behavioral factors that are linked to neuroticism, such as poor sleep, may lead to daytime sleepiness and, thus, a higher incidence of cognitive failures. Future research could include a measure of sleep quality to establish whether the link is directly between neuroticism and workplace cognitive failures or is more indirect through behavioral factors.

With respect to the environmental demands placed on everyday cognition, scores on the MPED ([43]) also had some predictive relationships with workplace cognitive failures. Busyness was found to be a weak positive predictor of WCFS total score and all three factors, while Routine showed very weak positive predictive relationships with WCFS total score and the WCFS Attention factor. There was, thus, some contribution of self-reported environmental demands to workplace cognitive failures (mainly in terms of perceived busyness). As noted in Section 1, Martin and Park define busyness in terms of density of obligations and routine as relating to the predictability of events that occur. Busyness will reflect the (often competing) environmental demands placed on a worker’s attentional resources both inside and outside the work setting, presenting more distractions to work task performance (see [10]; [35]). [20] ([20]) have highlighted the contributory roles of time pressure, interruptions, and multitasking in cognitive failures at work. The Job Demands—Resources Model (JD-R; [5] would argue that job demands (for example, the respondent’s self-perceived busyness) that exceed the worker’s cognitive resources would lead to cognitive failures in the workplace. The positive relationship found in the current study between busyness and workplace cognitive failures would be consistent with such an interpretation and explain its more diffuse influence relative to Routine. In relation to the weak positive relationships between Routine and the Attention factor, it would seem to highlight attention potentially wandering when environmental demands are following a familiar pattern, resulting in cognitive failures at work.

The current study focused on exploring the relationships between self-reported dyslexia traits and workplace cognitive failures in a community sample rather than in participants with formally diagnosed dyslexia. While this approach obviously limits the conclusions that can be drawn about the effects of officially diagnosed dyslexia on cognition in employment settings, it provides important insights into the relationship between dyslexia traits and workplace cognitive performance. With all online survey research, there are, of course, concerns around self-selection bias and not being able to fully describe the population from which the sample has been drawn (e.g., [2]; [23]). However, recruitment took place via Prolific, so that the population can be identified as those Prolific users meeting the current study’s screening criteria. Moreover, in studies comparing the data yielded by online data collection platforms, those obtained from Prolific users have been found to be ranked highly for the quality of the data that they yield ([18]; [52]).

Since the current study has indicated significant predictive relationships between dyslexia traits and workplace cognitive failures, it would be fruitful for future research to investigate workplace cognition in workers with diagnosed dyslexia, verified through the checking of educational psychologist’s reports. Doing so would allow for an even closer mapping between dyslexia-related cognitive difficulties and workplace cognitive performance to provide more direct evidence of the broader range of ways in which dyslexic workers need to be supported. While to dyslexia researchers the community sample approach is a clear limitation, it should be noted that it is less easy to gain access to workers with officially diagnosed dyslexia (in terms of both being able to recruit sample participants from this population and the opportunities for such participants to participate around their work and other personal commitments) than to students with the condition. Furthermore, it is important to understand the cognitive profile associated with dyslexia traits beyond the young adult range and in the specific everyday contexts (such as in employment settings) in which cognition is used. The current findings could help to highlight to employees and employers alike where formal diagnostic testing might be required to allow for official recognition of their condition and the subsequent provision of workplace support. Understanding the ways in which dyslexia traits can influence workplace cognition beyond reading and spelling difficulties is important in detecting underlying and undiagnosed neurodevelopmental problems in workers in jobs where literacy skills are not to the fore, but which nevertheless draw upon other aspects of cognition that are affected.

Self-report research can be criticized for the weakness of the correlations between self-reported everyday cognitive experiences and objective measures of cognition. However, this lack of correlation is hardly surprising given that objective measures are usually one-off tests of optimal performance that lack ecological validity and are measured in the range of minutes, while self-report measures tap experiences of typical levels of cognition over different tasks and over weeks or months (see [71]; [76], for more on this distinction between different levels of cognition). Furthermore, a recent meta-analysis by [27] ([27]), has indicated that there are relationships between scores on [7]’s ([7]) CFQ and at least some objective measures of executive function. It should also be noted that a range of other influencing variables (ADHD symptoms, age, mental wellbeing, busyness and routine) were taken into account in the current study, thereby strengthening the conclusions to be drawn about the predictive relationship between dyslexia traits and workplace cognitive failures. Moreover, the current study provides insights into how self-perceived dyslexia severity influences workplace cognition and highlights the potential importance to theory of considering individual differences in perceived (or objectively measured) severity of difficulties when investigating cognition in formally diagnosed dyslexics, rather than focusing solely on group differences.

With the exception of the ARQ ([69]) and the BFI-10 ([56]), all predictor variables had Acceptable to Good reliability coefficients in the current study. The Cronbach’s alpha value for the ARQ, however, fell in the Questionable range of [25]’s ([25]) classification indicating some concerns over the reliability of the measure. [72] ([72]) have recently validated an alternative non-diagnostic dyslexia screening tool for adults, the Adult Dyslexia Checklist ([68]), reporting higher validity and reliability than that reported for the ARQ. The authors also highlight, inter alia, issues over the validity and reliability of the ARQ. Future research might thus benefit from employing the Adult Dyslexia Checklist as a measure of dyslexia symptoms instead of the ARQ. The reliability coefficients obtained from the BFI-10 ([56]) are in line (or better than those reported in the literature (e.g., [57]).

The current study has focused on employees working in a central location in order to reduce possible variation in the chances of cognitive failures occurring when working in a regular workspace rather than in an environment in which both domestic and work activities occur and also fitted better with the broad type of in situ working environment for which the WCFS ([81]) was designed. Future research could, therefore, explore the experiences of dyslexic workers when working remotely (for a systematic review, see [8]) rather than in a central place of work.

Finally, the current study captured the workplace cognitive experiences of a predominantly White sample of UK workers. While the 87% of respondents who self-identified as being of white ethnicity is quite close to the England and Wales 2021 Census data indicating 81.7% of the total population as being white ([49]), future research on dyslexia (or dyslexia traits) in the workplace could aim to oversample respondents from minority groups in the UK (or elsewhere) to develop a more nuanced and intersectional understanding of their cognitive experiences at work.

The implications of a greater dyslexia-related self-perceived susceptibility to memory, attention, and action errors at work are twofold: firstly, such difficulties might be indicative of the need for formal assessment of dyslexia to identify the support needs for a worker who seems more frequently prone to cognitive failures but who has not yet been officially diagnosed and, secondly (by extension from adult dyslexia traits to confirmed dyslexia status), to be considered in support plans, workplace interventions (see [15], for a systematic review of workplace interventions for dyslexic workers), and the design and provision of assistive technology for workers with formal diagnoses of dyslexia.

## 5. Conclusions

In the current study, adult dyslexia traits were found to have significant positive predictive relationships with the self-reported frequency of workplace cognitive failures, after accounting for scores on a range of other variables. The findings provide important and specific insights into how dyslexia-related traits influence cognitive performance in the workplace, resulting in more frequent cognitive failures of memory, attention, and action in the workplace. Such difficulties present challenges both to dyslexia theory to explain and to employers to accommodate and support.

## Figures and Tables

**Figure 1 behavsci-15-01582-f001:**
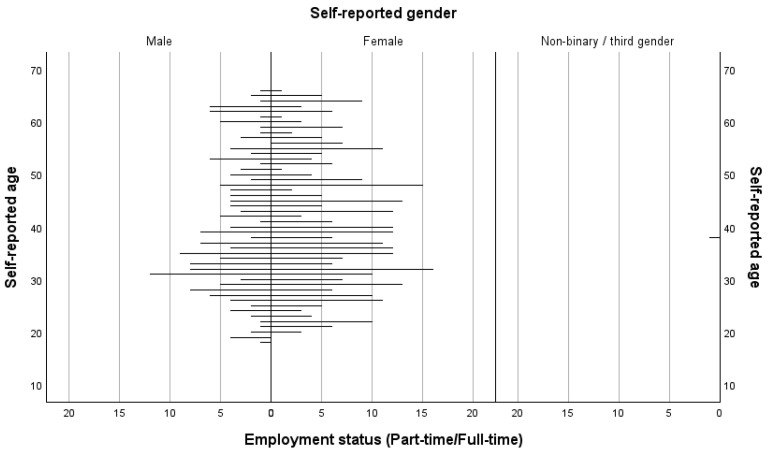
A participants pyramid showing the age and gender characteristics of the sample by employment status (part- and full-time).

**Table 1 behavsci-15-01582-t001:** Job sectors in which the respondents indicated that they worked (expressed as a percentage of the final sample).

Job Sector	Percentage
Banking, Accountancy, and Finance	3.5
Business Administration	9.8
Consulting and Management	2.3
Creative Arts and Design	2.5
Communications	0.8
Education	15.5
Healthcare	20.5
IT and Digital	3.3
Law	1.0
Other	41.0

**Table 2 behavsci-15-01582-t002:** Descriptive statistics and Cronbach’s α reliability coefficients for the self-report questionnaires.

Questionnaire Score	Mean (*SD*)	Cronbach’s α
MPED Busyness	20.03 (5.11)	0.844
MPED Routine	13.60 (2.96)	0.787
BFI-10 Extraversion	5.49 (2.18)	0.685
BFI-10 Agreeableness	7.16 (1.92)	0.576
BFI-10 Conscientiousness	7.53 (1.78)	0.565
BFI-10 Neuroticism	6.23 (2.31)	0.725
BFI-10 Openness	6.90 (1.92)	0.441
SWEMWBS score	23.50 (5.05)	0.879
ASRS score	9.37 (4.82)	0.837
ARQ score	10.92 (5.78)	0.675
**Workplace Cognitive Failures Scale**	**Mean (*SD*)**	**Cronbach’s α**
Total WCFS score	33.51 (9.83)	0.916
Memory factor	11.44 (3.75)	0.816
Attention factor	13.03 (4.27)	0.866
Action factor	9.05 (3.13)	0.797

**Table 3 behavsci-15-01582-t003:** Test statistics for the hierarchical regression analysis conducted on WCFS total score.

Predictor Variable	Model 1	Model 2	Model 3	Model 4	Model 5	Model 6
Age	−0.120 *	−0.082	0.005	0.013	0.055	0.050
MPED Busyness		0.378 ***	0.382 ***	0.350 ***	0.151 ***	0.171 ***
MPED Routine		−0.035	0.023	0.043	0.060	0.065 *
Extraversion			−0.073	−0.030	−0.010	0.005
Agreeableness			−0.096 *	−0.076	−0.027	−0.044
Conscientiousness			−0.391 ***	−0.320 ***	−0.159 ***	−0.157 ***
Neuroticism			0.113 **	0.066	0.068 *	0.061
Openness			0.071	0.066	−0.024	−0.005
SWEMBS				−0.201 ***	−0.032	0.002
ASRS					0.633 ***	0.483 ***
ARQ						0.247 ***
Adjusted-*R*^2^	0.012 *	0.157 ***	0.374 ***	0.397 ***	0.632 ***	0.667 ***
Δ*R*^2^	0.014 *	0.149 ***	0.223 ***	0.024 ***	0.231 ***	0.035 ***

Key: * *p* ≤ 0.05, ** *p* ≤ 0.01, *** *p* ≤ 0.001.

**Table 4 behavsci-15-01582-t004:** Hierarchical regression test statistics for the WCFS Memory factor.

Predictor Variable	Model 1	Model 2	Model 3	Model 4	Model 5	Model 6
Age	−0.150 **	−0.115 *	−0.044	−0.036	0.004	−0.001
MPED Busyness		0.358 ***	0.362 ***	0.330 ***	0.139 ***	0.160 ***
MPED Routine		−0.024	0.015	0.034	0.050	0.056
Extraversion			−0.124 **	−0.082	−0.063	−0.048
Agreeableness			−0.086	−0.066	−0.019	−0.037
Conscientiousness			−0.307 ***	−0.239 ***	−0.084 *	−0.082 *
Neuroticism			0.087	0.041	0.043	0.036
Openness			0.057	0.052	−0.034	−0.015
SWEMBS				−0.194 ***	−0.031	0.005
ASRS					0.610 ***	0.454 ***
ARQ						0.257 ***
Adjusted-*R*^2^	0.020 **	0.148 ***	0.300 ***	0.321 ***	0.540 ***	0.577 ***
Δ*R*^2^	0.023 **	0.132 ***	0.160 ***	0.022 ***	0.215 ***	0.037 ***

Key: * *p* ≤ 0.05, ** *p* ≤ 0.01, *** *p* ≤ 0.001.

**Table 5 behavsci-15-01582-t005:** Hierarchical regression test statistics for the WCFS Attention factor.

Predictor Variable	Model 1	Model 2	Model 3	Model 4	Model 5	Model 6
Age	−0.076	−0.046	0.046	0.052	0.091 **	0.087 **
MPED Busyness		0.314 ***	0.321***	0.296 ***	0.112 **	0.128 ***
MPED Routine		−0.015	0.048	0.063	0.079 *	0.084 *
Extraversion			−0.060	−0.026	−0.008	0.004
Agreeableness			−0.117 **	−0.101 *	−0.056	−0.069
Conscientiousness			−0.414 ***	−0.360 ***	−0.210 ***	−0.208 ***
Neuroticism			0.084	0.048	0.049	0.044
Openness			0.087*	0.084 *	0.001	0.015
SWEMBS				−0.156 **	0.001	0.028
ASRS					0.587 ***	0.468 ***
ARQ						0.196 ***
Adjusted-*R*^2^	0.003	0.099 ***	0.333 ***	0.346 ***	0.548 ***	0.569 ***
Δ*R*^2^	0.006	0.100 ***	0.240 ***	0.014 **	0.199 ***	0.022 ***

Key: * *p* ≤ 0.05, ** *p* ≤ 0.01, *** *p* ≤ 0.001.

**Table 6 behavsci-15-01582-t006:** Hierarchical regression test statistics for the WCFS Action factor.

Predictor Variable	Model 1	Model 2	Model 3	Model 4	Model 5	Model 6
Age	−0.092	−0.057	0.006	0.013	0.043	0.039
MPED Busyness		0.329 ***	0.328 ***	0.298 ***	0.155 ***	0.171 ***
MPED Routine		−0.059	−0.011	0.007	0.020	0.024
Extraversion			0.001	0.041	0.055	0.067
Agreeableness			−0.039	−0.021	0.014	0.001
Conscientiousness			−0.294 ***	−0.229 ***	−0.113 *	−0.111 *
Neuroticism			0.136 **	0.092	0.093 *	0.088 *
Openness			0.034	0.030	−0.035	−0.020
SWEMBS				−0.186***	−0.064	−0.037
ASRS					0.455 ***	0.333 ***
ARQ						0.201 ***
Adjusted-*R*^2^	0.006	0.122 ***	0.237 ***	0.256 ***	0.377 ***	0.399 ***
Δ*R*^2^	0.008	0.120 ***	0.124 ***	0.021 ***	0.120 ***	0.023 ***

Key: * *p* ≤ 0.05, ** *p* ≤ 0.01, *** *p* ≤ 0.001.

## Data Availability

The data that support the findings of this study are openly available in the Open Science Framework at https://osf.io/sbtxv, accessed on 1 September 2025.

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
