# Peer review of "Self-Reported Dyslexia Traits as Positive Predictors of Self-Reported Cognitive Failures in the Workplace"

_behavsci, 2025, doi:10.3390/bs15111582_

Round 1

Reviewer 1 Report

Comments and Suggestions for Authors

The chosen research topic is indeed very relevant, and the article meets all the requirements for such work. The article is properly substantiated by the work of other scientists, and the research methodology is clearly described. The research was conducted in accordance with research ethics and uses high-level statistical analysis. The results of the study are presented appropriately, and a scientific discussion and conclusions are provided. The article is suitable for publication. 

Author Response

We thank the reviewer for their positive response to the manuscript and their kind words.

Reviewer 2 Report

Comments and Suggestions for Authors

This is a successful and scientifically well-developed project. However, I believe it would be useful to highlight a few minor areas for improvement with a view to its final publication:

- Firstly, I feel that a clear statement of the study's objectives is missing at the end of the introduction section or, failing that, at the beginning of the methods section.
- It would be an excellent addition to include a participants pyramid graph to describe the sample.
- In the methods section, when describing the instruments used, reliability indicators (Cronbach's alpha or McDonald's omega) and validity indicators should be included.
- In the description of the measurement instruments, it would be a significant improvement to explain why each instrument was selected from among the different options available.
- It would be interesting to add a section at the end of the conclusions summarizing the main findings of the study.

Author Response

Comments 1: Firstly, I feel that a clear statement of the study's objectives is missing at the end of the introduction section or, failing that, at the beginning of the methods section.

Response 1: We thank the reviewer for this comment and have provided such a statement at the end of the Introduction, after the statement of the hypotheses.

Comments 2: It would be an excellent addition to include a participants pyramid graph to describe the sample.

Response 2: We thank the reviewer for this suggestion and have included a participants pyramid graph in the revised manuscript. The reviewer has not specified precisely what they would look to see represented in the figure, so we have split it by age, gender, and employment status (full- or part-time).

Comments 3: In the methods section, when describing the instruments used, reliability indicators (Cronbach's alpha or McDonald's omega) and validity indicators should be included.

Response 3: The reviewer is right to highlight the importance of reliability indicators. We have taken onboard the reviewer’s comment and added example Cronbach’s alpha values from the previous literature in the Method. As the reviewer is aware, we have already reported the Cronbach’s alpha values that we obtained for our study in Table 2 of the Results. We are reporting the use of already validated instruments but, as noted in the Discussion, the results are in line with similar but broader self-report measures of everyday cognition, thereby indicating good construct validity.

Comments 4: In the description of the measurement instruments, it would be a significant improvement to explain why each instrument was selected from among the different options available.

Response 4: The decision to use the ARQ and ASRS was based on the first author’s familiarity with these very well-established questionnaires from his previously published work (e.g., Protopapa & Smith-Spark, 2022; Christie et al., 2024). As the manuscript highlighted, there is a dearth of instruments available to assess workplace cognitive failures and the WCFS was chosen based on its pedigree of use by researchers. The decisions surrounding the measures (MPED, BFI-10, and SWEMWBS) used as statistical controls for other contributors to cognitive failure were made on the pragmatic basis of the instruments having good pedigrees but also having a small number of items. The SWEMWBS (which also had the benefit of focusing on functional aspects of mental health, as noted in the manuscript). This decision was made in order to retain the goodwill of participants by not producing an overly long survey. We are not aware of an alternative to the MPED to assess everyday busyness and routine. We have added a statement to this effect in Lines 245 to 249 of the Materials. We hope that this addresses the point of the reviewer satisfactorily. We feel that it would confuse the reader to name and discuss other questionnaires in the Method that were not actually used in the study and would also lead to a bloated Materials subsection. As the reviewer has already seen, where concerns about the instruments were identified during the study, these are considered in the Discussion (e.g., the ARQ and alternatives to it).

Comments 5: It would be interesting to add a section at the end of the conclusions summarizing the main findings of the study.

Response 5: We have added a brief Conclusion section to the manuscript.

We thank the reviewer for all their comments which have helped to strengthen the paper.

Reviewer 3 Report

Comments and Suggestions for Authors

Dear Authors,

I find this study interesting and valuable, with a manuscript that is easy to read and understand. However, I have some minor suggestions that may improve the readability and comprehensibility of the paper.

Introduction: Lines 108–135: I expect these measures to be explained and described in the Materials section. It is somewhat redundant and confusing when measures are detailed both in the Introduction and the Methods sections. I recommend shortening this part or moving some content to the Materials section.

Introduction: Lines 144 and 162: Please also name the concepts/constructs of the WCFS and CFQ. Generally, it would be easier to read the paper if you referred more often to the names of the concepts/constructs that are the focus, rather than just the questionnaire names. What is important are the concepts/constructs being measured, not the measures themselves. This is especially relevant for the Discussion section.

Procedure: Lines 291–292: Please provide more information about the “written debrief.” What was presented, and why was it needed? This is likely a good practice necessary for future studies, so please provide more details.

Results: Table 2: As you also use factors from the WCFS as criteria in regression analyses, Cronbach’s alphas for these factors should be reported.

Results: Below line 312: Please add correlations between predictors and criteria to the Supplementary Material and explain why you chose to include predictors in the regression models that are not significantly correlated with the criteria, if that is the case.

Results: Table 5: Is the Adjusted R² significant for the WCFS Attention factor? Please explain or indicate in all relevant places (supplement, table, and text) whether it is significant, if it was omitted by mistake.

Discussion: Lines 390–392: I am not sure which results you are referring to – regression models or correlation coefficients between predictors and criteria. However, this correlation table is not reported, and I recommend adding it to the Supplementary Material.

Discussion: Lines 396–411: This is the only "discussion" of the results. In the previous paragraph, the results were summarized, while in the following sections, limitations, recommendations, and implications were explained and discussed in detail. The study lacks a more thorough discussion of the results obtained. For example, why is only consciousness a significant predictor of cognitive failure? Why does dyslexia explain more unique variance for Memory than the other two factors? Why is ADHD a stronger predictor than dyslexia, etc.?

Discussion: Please add a brief conclusion at the end of the Discussion or in a separate section.

Best regards, and all the best with your paper and future research

Author Response

Comments 1: Introduction: Lines 108–135: I expect these measures to be explained and described in the Materials section. It is somewhat redundant and confusing when measures are detailed both in the Introduction and the Methods sections. I recommend shortening this part or moving some content to the Materials section.

Response 1: The information in the Introduction was written with the aim of providing an overview of the instruments in relation to the previous literature (and to make the hypotheses meaningful to the reader), while the Method was written to describe the actual administration and scoring of the instruments in detail. We have reviewed the content as recommended by the reviewer and moved appropriate content to the Materials section to avoid undue repetition and streamline the reporting.

Comments 2: Introduction: Lines 144 and 162: Please also name the concepts/constructs of the WCFS and CFQ. Generally, it would be easier to read the paper if you referred more often to the names of the concepts/constructs that are the focus, rather than just the questionnaire names. What is important are the concepts/constructs being measured, not the measures themselves. This is especially relevant for the Discussion section.

Response 2: We thank the reviewer for this point and have reviewed the Discussion and edited it where appropriate in line with the reviewer’s recommendation. Our approach had been to address the direct measures first and then broaden the point to the underlying constructs. We are not completely clear what the reviewer is requesting on Lines 144 and 162 but have tried to address the point by naming the WCFS factors (now on Lines 147-148) or describing the construct where it has not previously been described (now on Lines 165-166).

Comments 3: Procedure: Lines 291–292: Please provide more information about the “written debrief.” What was presented, and why was it needed? This is likely a good practice necessary for future studies, so please provide more details.

Response 3: Thank you for this comment. It is a standard ethical requirement for our university and for our country’s professional body (the British Psychological Society) to debrief participants at the end of the study and we followed these guidelines (as they were required as part of the ethical review process). As the survey was presented online, a written debrief was presented as the final page of the survey as we could not provide a verbal debrief in these circumstances. Given that debriefing participants is a commonly accepted practice in psychological research ethics, we would not want to overexplain it here but have rephrased the statement slightly and expanded upon it to give more detail. We hope that this response answers the reviewer’s point satisfactorily.

Comments 4: Results: Table 2: As you also use factors from the WCFS as criteria in regression analyses, Cronbach’s alphas for these factors should be reported.

Response 4: We thank the reviewer for this point and have added the details to the revised manuscript.

Comments 5: Results: Below line 312: Please add correlations between predictors and criteria to the Supplementary Material and explain why you chose to include predictors in the regression models that are not significantly correlated with the criteria, if that is the case.

Response 5: The study was pre-registered (as stated in the manuscript) and we said in the pre-registration that all the named factors would be entered into the hierarchical regression models. Therefore, they were entered to avoid deviating from the planned analyses. We have added a sentence to the manuscript to explain this. Table S1 was originally concerned with determining the intercorrelations between predictor variables and, thus, avoiding collinearity. To address the reviewer’s valid point about also including the predictors, we have added the correlations with the criterion variables to Table S1.

Comments 6: Results: Table 5: Is the Adjusted R² significant for the WCFS Attention factor? Please explain or indicate in all relevant places (supplement, table, and text) whether it is significant, if it was omitted by mistake.

Response 6: We thank the reviewer for their attentiveness in spotting this accidental omission (a cognitive failure on the part of the first author!). The Adjusted-R² is indeed highly significant and we have now indicated this level of significance in Table 5 and in the corresponding supplemental table.

Comment 7: Discussion: Lines 390–392: I am not sure which results you are referring to – regression models or correlation coefficients between predictors and criteria. However, this correlation table is not reported, and I recommend adding it to the Supplementary Material.

Response 7: We have attempted to further clarify the sentence, which followed directly on from the previous sentence in which we indicated that we were addressing the results of the WCFS factors in relation to dyslexia. We were talking about strength of relationships in relation to the regression models, so were referring to the standardized-β values (which are presented in the tables in the Results section). This ambiguity has been resolved in the revised version by stating clearly what values we are referring to in this context. We have also directed the reader back to the tables in the Results and, further to this, have also added the correlations with outcome variables to Table S1 (see Comments 5). To further facilitate the reading, we have moved what was the final sentence of the paragraph earlier in the paragraph, grouping it with other sentences dealing with dyslexia traits (and leaving their comparison with ADHD symptoms to the end of the paragraph).

Comments 8: Discussion: Lines 396–411: This is the only "discussion" of the results. In the previous paragraph, the results were summarized, while in the following sections, limitations, recommendations, and implications were explained and discussed in detail. The study lacks a more thorough discussion of the results obtained. For example, why is only consciousness a significant predictor of cognitive failure? Why does dyslexia explain more unique variance for Memory than the other two factors? Why is ADHD a stronger predictor than dyslexia, etc.?

Response 8: We are grateful to the reviewer for making this point. A more thorough discussion of the results is now provided.

Comments 9: Discussion: Please add a brief conclusion at the end of the Discussion or in a separate section.

Response 8: We have added a brief Conclusion section to the manuscript. We thank the reviewer for all their comments and feel that they strengthened the paper.